

# Evaluating soil salinity dynamics under drip irrigation in the Manas River Basin, Xinjiang: a long-term analysis (1996–2019)

Jianrong Shao*, Shuaihao Li*, Xiaohu Yang, Fenghua Zhang, Haichang Yang, Zicheng Peng and Tayyaba Zulfiqar

Agricultural College, Shihezi University, Shihezi, Xinjiang, China
* These authors contributed equally to this work.

## ABSTRACT

The Manas River Basin, located in Xinjiang, China, is one of the province's four major agricultural irrigation regions and the first in the country to implement large-scale drip irrigation. While drip irrigation has enhanced water use efficiency, it has also contributed to soil salinization, negatively impacting crop yields and soil health. This study examines the spatial and temporal evolution of soil salinity in the oasis area of the basin from 1996 to 2019. The study evaluates salinization dynamics under long-term irrigation practices using soil salinity inversion models, regression analysis, water-salt balance calculations, geostatistical techniques, and ArcGIS. The results reveal significant improvements in soil salinity conditions, with 78.02% of the region experiencing reduced salinity and 10.09% exhibiting deterioration. From 1996 to 2019, non-salinized soil increased by 1,403.46 km$^2$, mildly salinized soil expanded by 3,702.28 km$^2$, while saline soils decreased by 7,685.6 km$^2$. Statistical analysis indicates that soil salinity followed normal or logarithmic-normal distributions, with higher variability observed in 2016 and 2019. Despite these positive trends, challenges remain, particularly in the Shihezi, Manas, and Mosuowan irrigation zones, which still exhibit moderate to severe salinity. This study highlights the effectiveness of drip irrigation combined with improved management practices in mitigating soil salinity and enhancing soil quality. However, it emphasizes the need for targeted strategies to address residual salinization risks, ensuring sustainable agricultural development and ecological balance in arid regions.

## INTRODUCTION

Soil salinization is one of the most pressing ecological challenges globally, particularly in arid and semi-arid regions, where it is a prominent indicator of land degradation. Xinjiang, located in northwestern China, exemplifies this issue, with saline soil areas comprising approximately one-third of China's total saline land (*Hou et al., 2021*). This region, a major agricultural hub, particularly for cotton production, faces significant threats from soil salinization, which negatively impacts soil fertility and limits crop productivity (*Zhang et al., 2025*). The reduction in cotton yields due to increased salinity in the soils of Xinjiang

Corresponding author
Haichang Yang, yhc2012@126.com

has been a subject of considerable concern, as salinized soils impede cotton growth and undermine the long-term sustainability of agriculture in the region (*Yang et al., 2020*; *Zhang et al., 2024*). However, soil salinization in Xinjiang is exacerbated by irrigation practices, mainly using saline water to enhance soil moisture in an area characterized by scarce water resources. Cotton, known for its high salt tolerance, is often grown in areas where saline irrigation is practiced, and its ability to withstand moderate levels of salinity has been recognized (*Chaudhary et al., 2024*).

Furthermore, excessive salinization poses significant risks to crop yields, soil health, and agricultural productivity. Drip irrigation, widely adopted in arid regions as an efficient water-saving technology, has improved water use efficiency (*Yang et al., 2023*). However, when inadequate drainage systems are in place, salts accumulate in the soil as water infiltrates, creating a dynamic equilibrium between water infiltration, evaporation, and salt accumulation. This process complicates soil salinity management and requires integrated solutions to mitigate its impacts (*Farifteh et al., 2007*). The study aligns with the UN Sustainable Development Goals (SDGs), particularly SDG 2 (Zero Hunger) and SDG 15 (Life on Land). By reducing salinization and improving soil quality, it supports agricultural productivity and land sustainability, contributing to these global goals (*Lile, Ocnean & Balan, 2023*). Additionally, the technology and strategies developed could be applied to other arid regions, such as Central Asia, North Africa, and the Middle East, to address similar salinity challenges.

Recent literature has explored soil salinity's temporal and spatial variations using various techniques. For instance, researchers have utilized Kriging, Bayesian maximum entropy methods, and artificial neural networks to understand the distribution of soil salinity (*Wang, Pan & Luo, 2019*). While these methods have provided valuable insights, they are limited by their inability to accurately capture the complex dynamics of soil salinity, mainly when applied in isolation. Single-method approaches often fail to integrate the multifaceted nature of soil salinization, resulting in suboptimal predictions and limited practical applicability in real-world scenarios (*He et al., 2023*). Building on these previous studies, this research aims to advance the understanding of soil salinity dynamics by employing a combined model that integrates multiple methodologies to better capture the spatial and temporal evolution of soil salinity in the Manas River Basin, Xinjiang, over a period spanning from 1996 to 2019. This study utilizes advanced techniques such as soil salinity inversion models, regression analysis, water-salt balance calculations, geostatistical methods, and ArcGIS to assess the region's long-term trends of soil salinization. The research will quantify how these practices have transformed salinization levels, from severe to moderate or mild salinity, by analyzing the impacts of drip irrigation and associated management practices over the last two decades. The findings are expected to highlight the effectiveness of these interventions in reducing the extent of salinized soils, particularly in areas such as Shihezi, Manas, and Mosuowan irrigation zones.

This study also introduces the concept of a critical ratio of drainage to irrigation (CRDI), a novel metric developed to optimize irrigation and drainage practices in the region. The CRDI aims to maintain the balance between water application and salt leaching, preventing the accumulation of salts in the soil matrix and promoting long-term

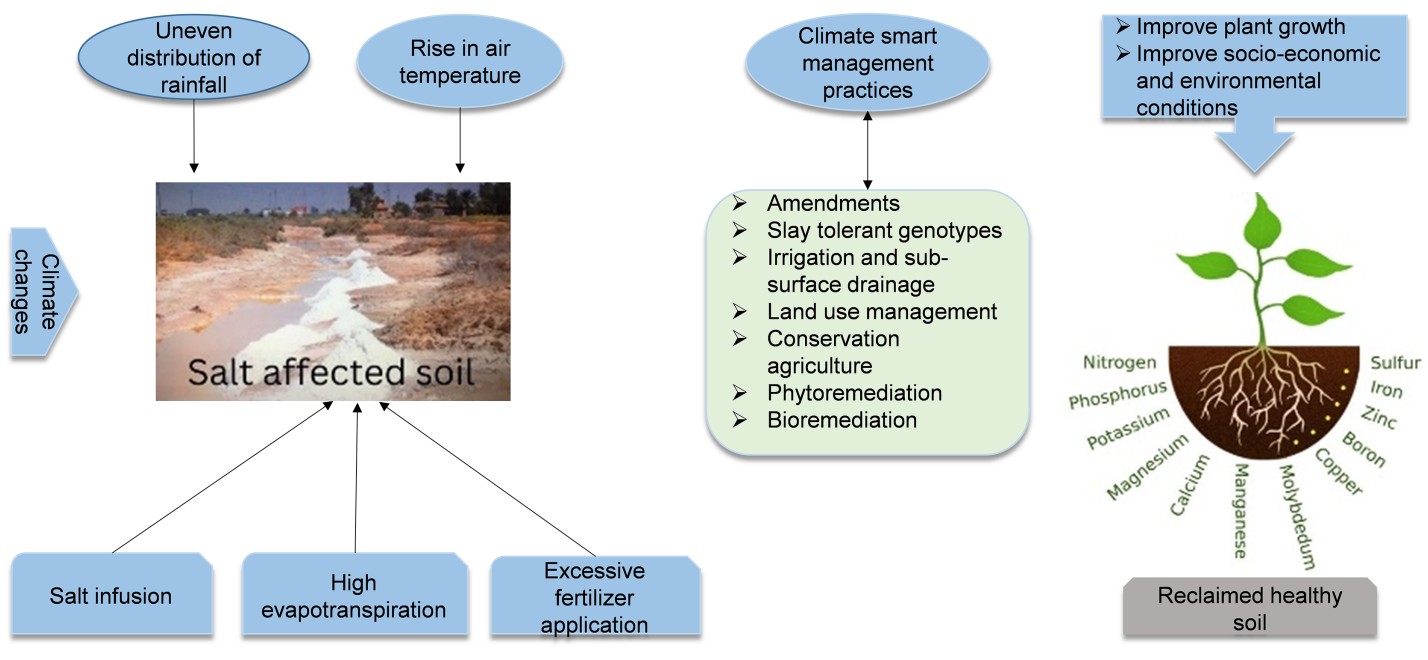

**Figure 1** Climate change and soil salinity: obstacles to sustainable agriculture and food security.

soil health. By quantifying the relationship between soil salinity, groundwater depth, irrigation drainage, and land use change, this research provides a comprehensive framework for managing soil salinity in the Manas River Basin. The results will inform strategies for sustainable agricultural development and ecological stability in arid and semi-arid regions, providing a solid scientific foundation for mitigating the risks of soil salinization and supporting the continued growth of cotton and other vital crops (Fig. 1). Ultimately, this research will contribute to formulating policies and management practices to sustain agricultural productivity and foster environmental resilience in Xinjiang and similar arid regions worldwide.

## MATERIALS AND METHODS

### Study area

The Manas River Basin is located in the hinterland of the Eurasian continent, on the edge of the Gurbantungut Desert, the largest fixed and semi-fixed desert in China (84°43′–86°35′E, 43°21′–45°20′N) (*Yang et al., 2020*). The Xiayedi irrigation area, Anjihai irrigation area, Jingouhe irrigation area, Shihezi irrigation area, Xinhuchang irrigation area, Mosuowan irrigation area, and Manas River irrigation area are the seven irrigation regions in the basin. From south to north, the basin's primary geomorphologic features—typical mountainous oasis basin ecosystems—include mountains, piedmont folds, alluvial fans, plains, dry deltas, and deserts. The average temperature in the basin was 6.5 °C. The highest temperature in the year was in July, and the lowest was in January. The frost-free period was 155 d. The sunshine duration was 2745 h. The annual precipitation was 125.0–207.7 mm, and the annual evaporation was 1,000–1,500 mm (2000–2017).

## Soil sample collection and treatment

Field sampling and investigation were conducted at the Manas River Basin Oasis in October 2019. The sampling points were chosen based on their homogeneity, representativeness, salinization state, and surface characteristics in the study area. According to the five-point sampling approach, soil samples were collected in the 30 m by 30 m quadrat at 0 to 20 cm depth. Three hundred fifty effective sampling points were produced (Fig. 2) after precisely determined each sampling point's longitude and latitude. Three hundred fifty soil samples were air-dried, crushed, and sieved through a 2 mm-aperture soil before being uniformly combined. A total of 200 g soil samples were collected using the quartering method. The electrical conductivity was measured, and soil salinity (SSC, g/kg) was determined using an empirical formula (*Pouladi et al., 2019*). As shown in Eq. (1):

$$SSC = (2.16EC_{5:1} + 2.45). \tag{1}$$

In Eq. (1), EC5:1 is the soil conductivity measured by soil water ratio 1:5.

The soil salinization in the study area is divided into five grades. The grading standard of soil salinization used in this study is shown in Table 1.

## Remote sensing data acquisition

This study used six distinct years' worth of NASA's Landsat series remote sensing photos, considering sampling time and remote sensing image quality.

## Geometric correction

Remote sensing image deformation is generally caused by systematic and non-systematic two reasons (*Roth et al., 2018*). The influence of latitude, height, and external environment of different sensor platforms is called systematic influence; remote sensing images with uneven rows, different pixel sizes, and irregular shapes are called non-systematic effects . Geometric correction is usually used for non-systematic. The calibrated remote sensing image is chosen for geometric correction in this work. The ENVI 5.1 toolbox is used to choose the control points. The notion of prominent and simple-to-identify sites, such as typical buildings, river bends, road intersections, and so forth, is followed in selecting control points. The polynomial algorithm geometrically correct the remote sensing images after uniformly distributing as many control points as possible.

## Image clipping

Cutting remote sensing images according to the boundary of the study area can effectively reduce the amount of data and improve the operation speed. In this study, ENVI 5.1 is used to cut the boundary of the study area, and the remote sensing image of the study area in 2019 is obtained.

## Radiometric calibration

The foundation for achieving remote sensing information quantification is radiometric calibration, which involves converting distinct remote sensing images into radiance values (DN) using an atmospheric radiative transfer model (*Xu et al., 2019*). Based on ENVI 5.1,

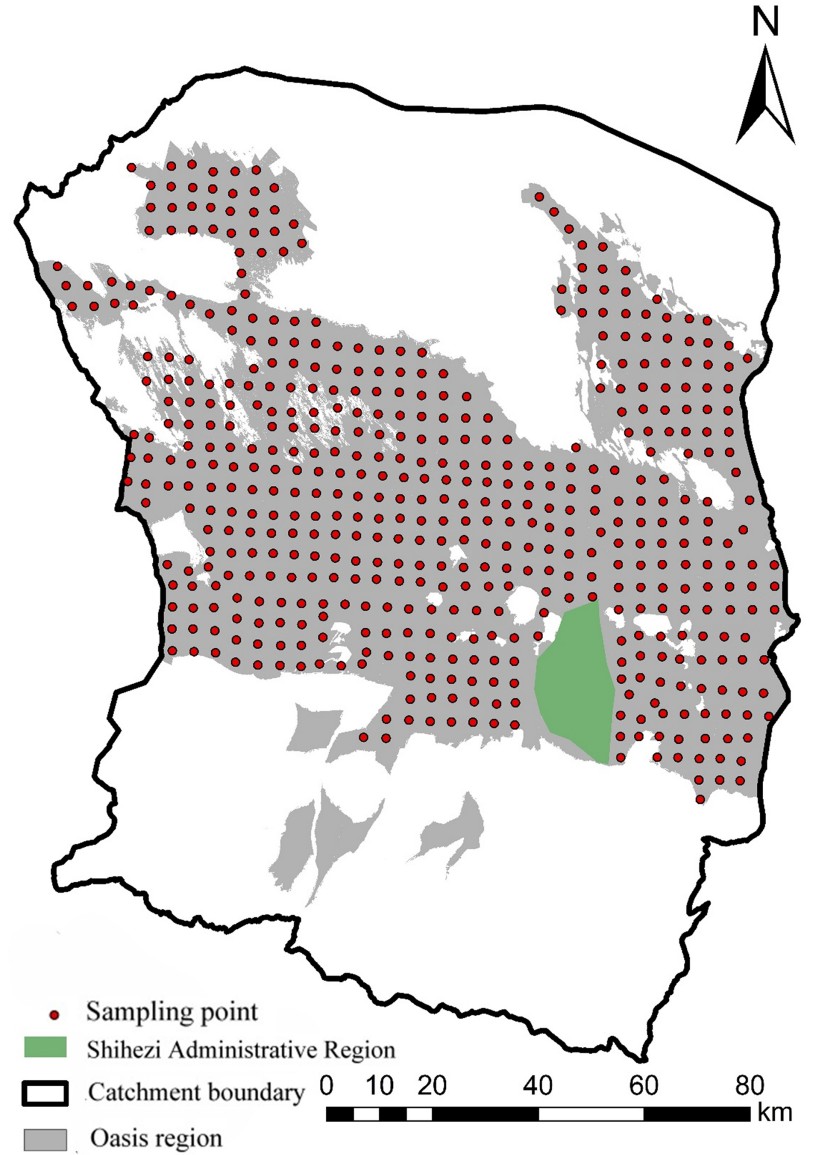

**Figure 2 Distribution map of sampling points in the oasis of the Manas River Basin.**

**Table 1 Statistics of available soil salinity.**

| Dataset | Number of samples | Mean (g/kg) | Minimum (g/kg) | Maximum (g/kg) | SD (g/kg) | CV (%) |
|---|---|---|---|---|---|---|
| Total sample | 350 | 4.59 | 41.25 | 0.30 | 7.44 | 61.81 |
| Modeling sample | 280 | 4.84 | 41.25 | 0.30 | 7.79 | 62.13 |
| Validation sample | 70 | 4.08 | 39.10 | 0.31 | 6.63 | 61.48 |

the visible-near-infrared data is chosen for this work, and the calibration type is radiation brightness value. The output type is float, while the storage form is BIL. Radiation brightness has a unit conversion coefficient of 0.1. The radiometric calibration of the photos from remote sensing is finally finished.

## Flash atmospheric correction

The true reflectance of ground objects is determined by sensors using atmospheric correction which simultaneously supports a range of sensor types and has high accuracy and robust autonomy. Based on ENVI 5.1, this study performs atmospheric correction using radiometric calibration data, determines the actual reflectance of ground objects, and completes the Flash atmospheric correction of remote sensing photos.

## Supervision classification

Remote sensing technology plays a vital role in land-use and land-cover classification, enabling spatial information extraction for environmental monitoring and resource management. Supervised classification is widely used among various classification techniques because it can categorize unknown pixels based on known training samples (*Luo, 2019*). This study employed supervised classification to analyze remote sensing images and generate land-use information for the Manas River Basin. Using ENVI 5.1 software, training areas representing distinct land-use types were selected to classify satellite imagery, ensuring accurate classification results. The generated thematic map provides valuable insights into the spatial distribution of oasis land-use types, contributing to a better understanding of land-use dynamics in the study area.

## Calculation of remote sensing index

Nine multispectral remote sensing indicators closely connected to soil salinity were chosen for this study. Salinity index (SI, $SI_1$, $SI_2$, $SI_3$), ratio vegetation index (RVI), soil adjusted vegetation index (SAVI), normalized difference water index (NDWI), normalized difference vegetation index (NDVI), difference vegetation index (DVI), normalized difference vegetation index (NDVI), (*Meivel & Maheswari, 2022*). Calculations are made using ENVI 5.1's Band math toolkit. In Table 2, the computation formula is displayed.

## Data analysis and validation

In this study, 350 soil samples were randomly divided into two groups using the sampling data from 2019. One group of 280 samples was used for modeling, and the other 70 samples were used for verification.

The weighted average approach is utilized to give a bigger weight to the model with high verification accuracy and a lower weight to the model with low verification accuracy. The variable weight combination prediction method is chosen. Calculation approach as demonstrated by Eqs. (2) and (3):

$$W_i = \frac{\sigma - \sigma_i}{\sigma} \times \frac{1}{n-1} \tag{2}$$

**Table 2 Statistic of soil salinization area at different levels.**

| Degree of salinization | Area (km$^2$) | | | | | |
|---|---|---|---|---|---|---|
| | 1996 | 2002 | 2008 | 2012 | 2016 | 2019 |
| Non-salinization | 471.93 | 1,223.44 | 2,205.65 | 1,282.81 | 1,241.35 | 1,875.39 |
| Mild salinization | 72.38 | 903.27 | 1,060.76 | 1,575.77 | 3,003.93 | 3,774.66 |
| Moderate salinization | 101.33 | 913.23 | 1,304.71 | 2,630.57 | 1,648.28 | 1,508.35 |
| Severe salinization | 509.25 | 1,276.08 | 1,856.32 | 2,557.58 | 2,336.77 | 1,681.06 |
| Saline soil | 8,179.82 | 5,018.68 | 2,907.26 | 1,287.98 | 1,101.43 | 494.22 |

$$\sigma = \sum_{i=1}^{n} \sigma_i \quad i = 1, 2, \ldots n. \tag{3}$$

In the formula, $W_i$ is the weight of model $i$, $\sigma_i$ is the standard deviation of the prediction error of model $i$, and $n$ is the number of single models.

The determination coefficient ($R^2$) and root mean square error (RMSE) are the evaluation indices of the model's stability and prediction accuracy. Strong prediction ability and great stability are characteristics of the model with a large determination coefficient and a modest RMSE. As shown by each test phrase in Eqs. (4) and (5),

$$R^2 = \frac{\sum_{i=1}^{n} X_i - \bar{X} \times (Y_i - \bar{Y})}{\sqrt{\sum_{i=1}^{n} (X_i - \bar{X})^2 \times \sum_{i=1}^{n} (Y_i - \bar{Y})^2}} \tag{4}$$

$$RMSE = \sqrt{\frac{\sum_{i=1}^{n} Y_i^0 - Y_i^{s\,2}}{n}} \tag{5}$$

where: n is the number of sampling points; $X_i$ is the measured value of soil sample I; $\bar{X}$ is the average of measured values of soil samples; $Y_i$ is the predicted value of soil sample i; $\bar{Y}$ is the average value of soil sample prediction; $Y_i^0$ and $Y_i^s$ are the predicted and real values of soil samples (Chicco, Warrens & Jurman, 2021).

## RESULTS

### Soil salinity inversion in the Manas River Basin Oasis, 1996–2019

The spatial distribution of soil surface salinization in six periods from 1996 to 2019 (Fig.3). The majority of the surface soil in the entire basin had significant salinization and saline soil in 1996, and the level of salinization was severe, as can be seen from the map. Following the initial application of large-area drip irrigation in 2002, soil salinization in the study region was much lower than it was in 1996. As of 2008, to accomplish complete drip irrigation, soil salinization had been further facilitated, and soil salinity had considerably lowered after 12 years of film drip irrigation technology combined with acceptable fertilization and other enhanced saline-alkali soil technology. In 2012, there was a clear upward trend in the salt content of surface soil, and from that year until 2019, there was a clear decrease trend.

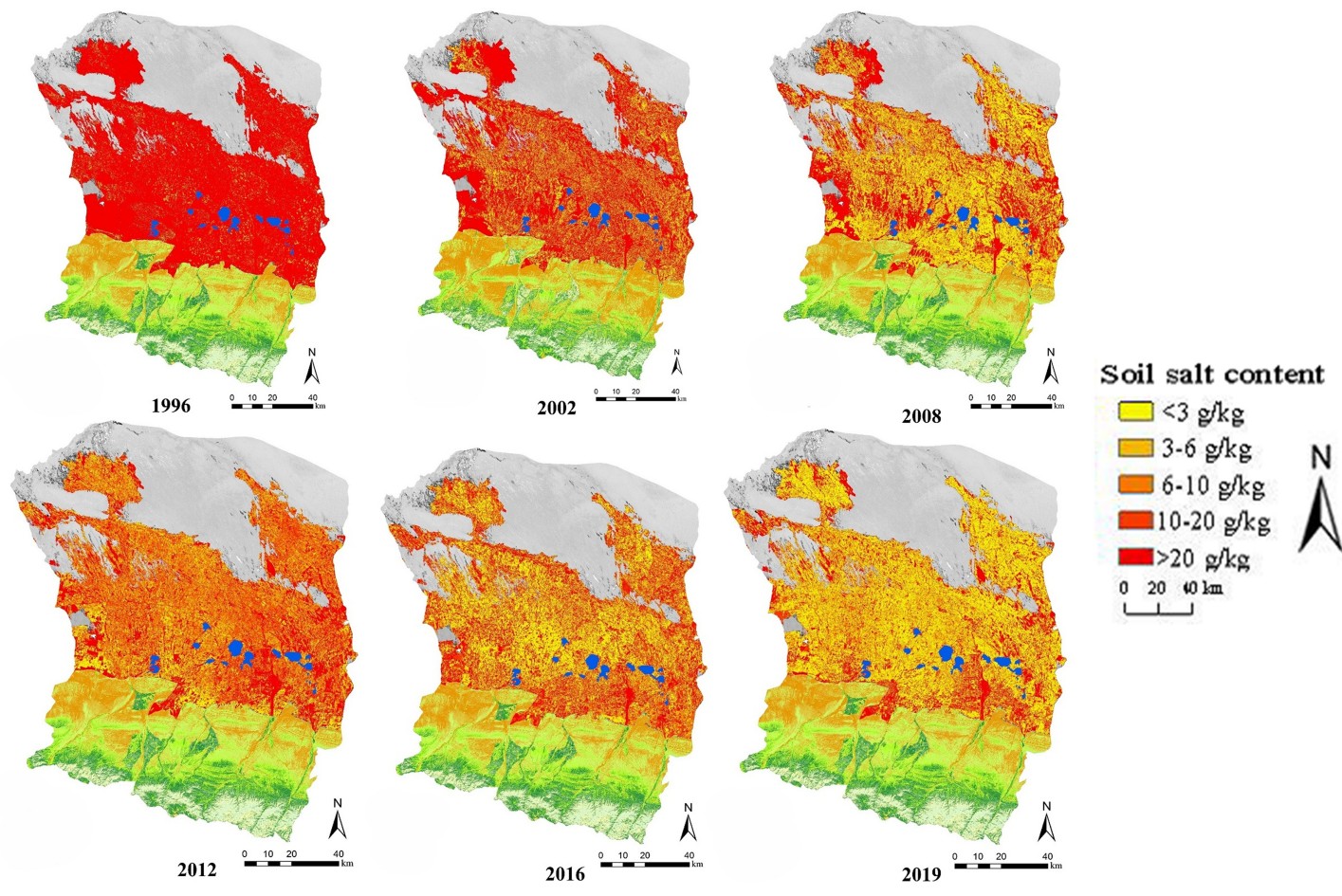

**Figure 3 Spatial distribution map of soil salinity from 1996 to 2019.**

## Descriptive statistical analysis of soil salt content in different years

It is clear from Table 2 that the soil salt content in 1996 and 2002 was much higher than that in previous years. This is based on the average and standard deviation of soil salt content across various years. In 2016 and 2019, the soil salinity showed significant variance, with coefficients of variation of 128.64% and 127.92%, respectively. In 1996, 2002, 2008, and 2012, the coefficients of variation of soil salinity were, reasonably modest and corresponded to the range of medium fluctuation.

Figure 4 shows how drastically varied the distribution of soil salt is between various years. In 2008, 2012, 2016, and 2019, there was a noticeable skewed distribution of soil salinity to the left. In 2008 and 2019, the surface soil salinity had a higher skewness coefficient, and the kurtosis values were 3.36 and 2.86, respectively, deviating from the normal distribution. The variation in soil salinity in 1996, 2002, 2012, and 2016 was non-significant, and the kurtosis values, which complied with the normal distribution, were 0.29, 0.46, 0.76, and 0.39, respectively. The soil salt content from 1996 to 2019 was tested using the K–S normal test, and the results ($p \leq 0.05$) showed that the soil salt level in 1996

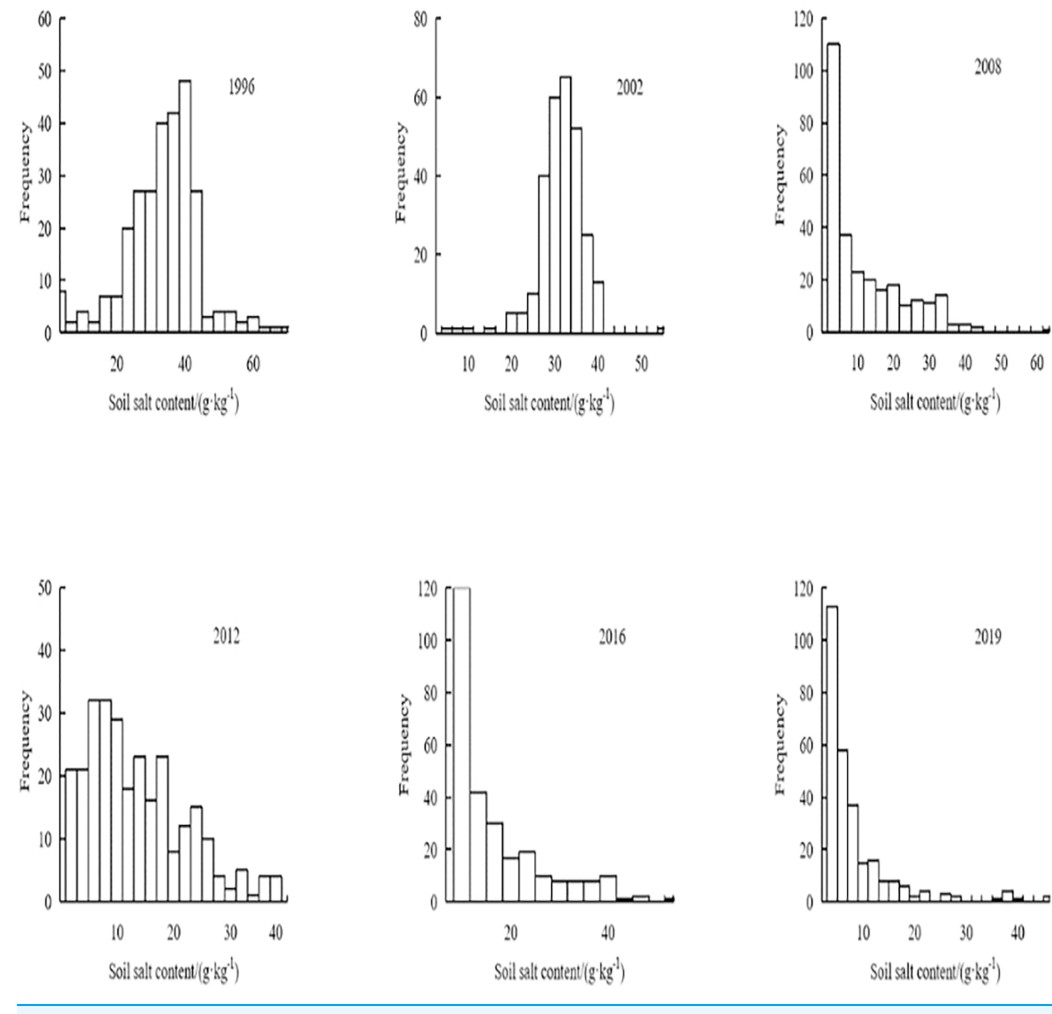

**Figure 4 Frequency histogram of soil salt contents.**

and 2002 followed the normal distribution, while the soil salt content in 2008, 2012, 2016, and 2019 followed the logarithmic normal distribution.

## Spatial-temporal evolution of soil salinity in Manas River Basin

According to the statistical findings, between 1996 and 2019, when drip irrigation under mulch was first introduced, the area of non-salinized soil increased by 1,403.46 km², the area of mildly salinized soil increased by 3,702.28 km², and the area of saline soil decreased by 7,685.6 km² in the study area . The transition of saline soil into severe and moderate salinization, and moderate and severe salinization into mild and non-saline soil, intensified salinization. When drip irrigation under mulch first gained popularity, the study area's saline soil gradually changed to moderately and severely salinized soil. As a result, the salinization phenomenon was significantly reduced, but the moderately and severely salinized area increased from 1996 to 2019. After mulched drip irrigation became widely used, mild soil salinization was achieved using of flawless management technology.
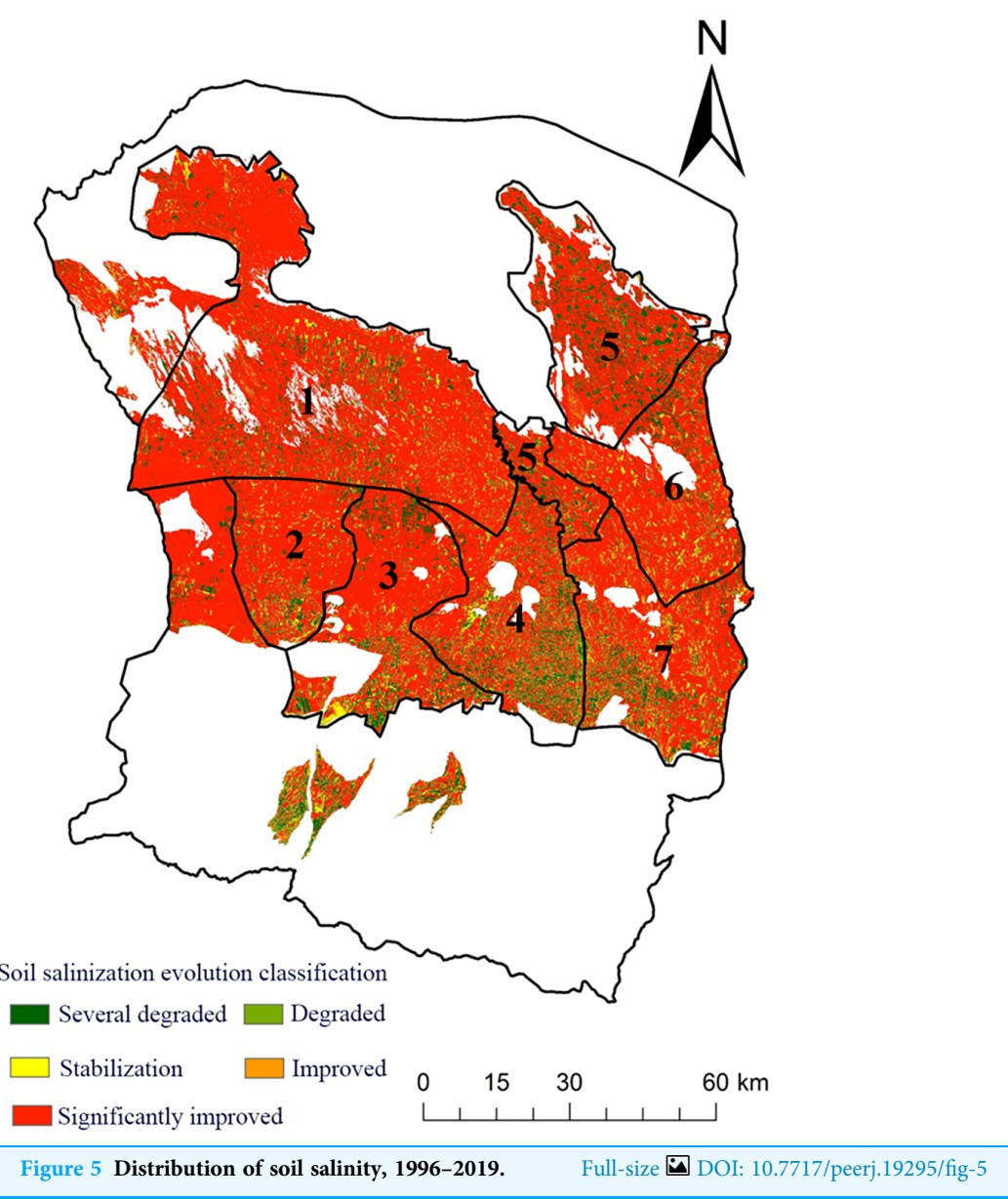

**Figure 5  Distribution of soil salinity, 1996–2019.**   

However, a significant amount of soil in 2019 is still moderately or severely salinized, and possible dangers could limit agricultural development and sustainable land use. Soil salinization management needs to be strengthened to secure future economic growth and create a healthy ecological environment.

## Trend analysis of soil salinization

This article uses ArcGIS 10.2 software to calculate the difference in salt content between 1996 and 2019 to further explore the spatial evolution of the oasis in the Manas River Basin over the past 23 years. The evolution results are divided into five categories: noticeable improvement, stability, deterioration, and apparent deterioration. The specifics of the

**Table 3 Correlation analysis of soil salinity and spectral index.**

|  | SSC | SAVI | RVI | SI | SI$_1$ | SI$_2$ | SI$_3$ | NDVI | NDWI | DVI |
|---|---|---|---|---|---|---|---|---|---|---|
| SSC | 1 | | | | | | | | | |
| SAVI | −0.733** | 1 | | | | | | | | |
| RVI | −0.663** | 0.947** | 1 | | | | | | | |
| SI | 0.640** | −0.665** | −0.776** | 1 | | | | | | |
| SI$_1$ | 0.664** | −0.680** | −0.789** | 0.984** | 1 | | | | | |
| SI$_2$ | 0.354** | −0.174** | −0.368** | 0.817** | 0.834** | 1 | | | | |
| SI$_3$ | 0.663** | −0.680** | −0.791** | 0.983** | 0.999** | 0.836** | 1 | | | |
| NDVI | −0.746** | 0.967** | 0.985** | −0.802** | −0.820** | −0.393** | −0.820** | 1 | | |
| NDWI | 0.742** | −0.935** | −0.923** | 0.801** | 0.804** | 0.359** | 0.798** | −0.951** | 1 | |
| DVI | −0.680** | 0.975** | 0.864** | −0.507** | −0.519** | 0.031 | −0.518** | 0.887** | −0.869** | 1 |

Notes:
SSC is soil salt content.
**Extremely significant correlation.

research area's results for the evolution of soil salinization are displayed in the statistics table. Overall, there was a noticeable improvement in soil salinization from 1996 to 2019; the soil area with a substantial improvement in salinization reached 7,282.28 km$^2$, or 78.02% of the research region's total area. The area where soil salinization improved was 238.07 km$^2$, making up 3.62% of the study area's total area; the area where soil salinization remained stable was 620 km$^2$, making up 6.64% of the study area's total area; and the area where soil salinization declined was 151 km$^2$, making up 1.62% of the study area's total area. The area of the study region's soil that had deteriorated due to salinization was 942.10 km$^2$, or 10.09% of the total area.

The distribution map of soil salinity evolution in the study area from 1996 to 2019 is shown in Fig. 5. The Shihezi irrigation area, Manas irrigation area, and Mosuowan irrigation area all exhibit more pronounced deterioration tendencies than the study area. Overall, the study area's soil salinization has been significantly reduced; some saline soil has transformed into severe and moderate salinization soil, and some moderate and severe salinization soil has transformed into mild and non-salinized soil, indicating that, over the past 23 years, soil salinization has only intensified in a small portion of the study area and is still improving overall. In addition to improving soil quality and speeding up the process of self-healing, perennial drip irrigation conditions have helped to minimize the creation of soil salinization to some level.

## Correlation analysis of spectral indices with soil salt concentration

The correlation analysis confirms the relationship between spectral indices and soil salt concentration (SSC), with SI, SI1, and SI3 positively correlated with salinity, while NDVI and SAVI show strong negative correlations (Table 3). These findings validate the selected indices' effectiveness in detecting soil salinity patterns. The correlation results are presented in the Results section to align with their analytical (*Ning et al., 2021*; *Qi et al., 2022*).

## DISCUSSION

The Manas River Basin oasis region, characterized by substantial salt deposits, experiences soil salinization due to natural and anthropogenic factors. Mulched drip irrigation, a widely adopted water-saving technology, has sparked debate regarding its role in secondary salinization. In contrast, some studies suggest that mulched drip irrigation retains soil salts and potentially increases salinized areas over time (*Yang et al., 2019*); Others argue that a high water table and excessive evaporation relative to rainfall drive secondary salinization. This process results in salt accumulation at the soil surface, leading to long-term soil degradation. However, this study's numerical modeling and field measurements indicate that salt distribution and migration under mulched drip irrigation differ from conventional secondary salinization processes. Drip irrigation, particularly when integrated with precise water and fertilizer management, enhances soil water infiltration and reduces salt buildup. *Ren et al. (2019)* highlighted that irrigation schedules are critical in maintaining soil moisture and controlling salinity in desert environments. Previous studies by *Wang et al. (2024)* and *Hou et al. (2019)* demonstrate that high-frequency drip irrigation replenishes moisture in shallow and deep soil layers, mitigating evaporation losses and ensuring consistent soil hydration. This study showed that the irrigation measures considerably improved the soil water infiltration performance because the water content of the soil layers changed significantly over time under various irrigation patterns and gradually rose with increasing soil depth. The study found that medium and large volumes of drip irrigation significantly increased the soil water content in both shallow (0–20 cm) and deep (70–100 cm) soils compared to traditional flood irrigation. This suggests that high-frequency drip irrigation helped replenish the waters lost in shallow soils due to high evaporation and transpiration in arid and semi-arid areas, thereby maintaining high surface soil moisture levels (*Dong et al., 2022*).

Consequently, the primary cause of the variation in soil water content between the two treatments may be the substantial volume of low-frequency irrigation water treated by flood irrigation and the high-frequency uniform irrigation water treated by drip irrigation. Under drip irrigation, the consistent and steady infiltration of irrigation water boosted the water content of deep soil and preserved the high soil water content in the high evaporation climate. When forecasting soil salinity or EC, it is important to take into account the impact of soil texture, soil moisture, and irrigation (*Wang et al., 2022*). Conventional laboratory analysis, which is frequently complex, costly, time-consuming, and arduous, is necessary to assess soil salinity accurately (*Xiao et al., 2023*). Process-based models require extensive expertise and complex input data for accurate predictions, whereas multispectral data from UAVs or optical remote sensing satellites offers a more accessible and efficient alternative for monitoring and analyzing soil salinity (*Gorji et al., 2020*). Recent advancements have made these techniques more precise and easier to apply. More research has employed process-based and remote sensing models to simulate soil salinity dynamics in cotton fields (*Ivushkin et al., 2017*; *Ning et al., 2021*), providing valuable recommendations for irrigation scheduling and water-salt balance management.
Additionally, the growing use of machine learning (ML) has gained prominence due to the availability of large datasets enabled by remote sensing technologies. Remote sensing technologies, including UAVs and satellite imagery, provide an efficient alternative for monitoring and analyzing soil salinity. Recent advancements in machine learning (ML) and spectral fusion techniques have improved the precision of soil salinity mapping (*Qi et al., 2022*). For instance, ML-driven approaches using remote sensing data have successfully simulated salinity dynamics and provided actionable recommendations for irrigation schedules and water-salt balance management (*Ning et al., 2021*). The NDVI was selected because soil salinity directly impacts vegetation health, leading to reduced chlorophyll content and plant cover, which lowers NDVI values. The strong negative correlation between NDVI and soil salinity ($r = -0.746$, $p < 0.01$, Table 2) confirms its effectiveness as an indirect salinity indicator. Previous studies (*Yang et al., 2020*; *Zhang et al., 2011*) have demonstrated that NDVI can differentiate between saline and non-saline areas by detecting vegetation stress, making it highly relevant in oasis ecosystems like the Manas River Basin. SWIR-based indices were not prioritized due to data consistency, mixed-pixel effects, and land cover heterogeneity. The study spans 1996 to 2019, and SWIR band availability varies across Landsat missions, limiting comparability. Additionally, SWIR indices are highly sensitive to moisture and soil texture, leading to potential misclassification in the Manas River Basin, where irrigated fields, saline soils, and desert interact. VNIR-based indices (SI, NDVI, and NDWI) showed strong empirical correlations with soil salinity (Table 3), supporting their selection (*Farifteh et al., 2007*; *Gorji et al., 2020*). This study's findings align with prior research, showing reduced soil salinity under long-term drip irrigation and mulch application from 1996 to 2019. However, discrepancies in prior studies regarding salt accumulation under mulched drip irrigation may stem from methodological limitations, such as comparing salt content across different farmlands rather than within the same field. Future research should adopt more robust methodologies to address these inconsistencies and provide deeper insights into the impacts of drip irrigation on soil salinity.

## CONCLUSIONS

Remote sensing imagery provides valuable spatial details, enabling the derivation of indices to characterize soil salinity, even in un-sampled areas. A combined model analyzed the regional and temporal evolution of soil salinity in the study area over 6 years from 1996 to 2019. Results indicate that soil salinity was significantly higher in 1996 and 2002 compared to later years. Over the study period, salinization decreased notably, with substantial variability observed in 2016 and 2019, while earlier years (1996, 2002, 2008, and 2012) showed more stable trends.

Despite this progress, a considerable portion of the region remains moderately to severely salinized, posing challenges to crop productivity and sustainable land use. Notably, 78.02% of the study area experienced reduced salinization from 1996 to 2019. Adopting drip irrigation and preventive management practices has effectively curbed salinization and improved soil quality. However, persistent salinity in specific zones

highlights the need for enhanced strategies to mitigate salinization risks and ensure sustainable agricultural development.

### Funding

The National Natural Science Foundation of China (42167036), the project was funded by the Corps Science and Technology Plan Project (2022ZD055), the Key Research and Development Plan for the Autonomous Region (2022B02020-1), and the Chinese International Science and Technology Cooperation Base "Oasis's Crops Efficient Production and Agricultural Environmental Protection". The funders had no role in study design, data collection and analysis, decision to publish, or preparation of the manuscript.

### Grant Disclosures

The following grant information was disclosed by the authors:
The National Natural Science Foundation of China: 42167036.
Corps Science and Technology Plan Project: 2022ZD055.
Key Research and Development Plan for the Autonomous Region: 2022B02020-1.
Chinese International Science and Technology Cooperation Base "Oasis's Crops Efficient Production and Agricultural Environmental Protection".

### Competing Interests

The authors declare that they have no competing interests.

### Author Contributions

- Jianrong Shao conceived and designed the experiments, performed the experiments, analyzed the data, prepared figures and/or tables, authored or reviewed drafts of the article, and approved the final draft.
- Shuaihao Li conceived and designed the experiments, performed the experiments, analyzed the data, prepared figures and/or tables, authored or reviewed drafts of the article, and approved the final draft.
- Xiaohu Yang performed the experiments, prepared figures and/or tables, and approved the final draft.
- Fenghua Zhang conceived and designed the experiments, authored or reviewed drafts of the article, and approved the final draft.
- Haichang Yang conceived and designed the experiments, authored or reviewed drafts of the article, and approved the final draft.
- Zicheng Peng performed the experiments, analyzed the data, prepared figures and/or tables, and approved the final draft.
- Tayyaba Zulfiqar performed the experiments, prepared figures and/or tables, and approved the final draft.

## Data Availability

The raw data is available in the Supplemental File.

## Supplemental Information

Supplemental information for this article can be found online at http://dx.doi.org/10.7717/peerj.19295#supplemental-information.

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
