# Peer review of "Evaluating soil salinity dynamics under drip irrigation in the Manas River Basin, Xinjiang: a long-term analysis (1996–2019)"

_PeerJ, doi:10.7717/peerj.19295_

## Round 0.1 · original submission · Major Revisions

*Abstract: Rewrite the abstract to ensure clarity and completeness by including all key components of the study.
*Introduction: Strengthen the introduction by incorporating recent literature and providing a clearer contextualization of the study's significance.
*Methodology: Justify key methodological choices, such as the use of supervised classification and specific spectral indices.
*Data Preprocessing: Include detailed descriptions of the data preprocessing steps to improve transparency and reproducibility.
*Model Details: Clearly explain the model variables, calibration processes, and validation procedures used in the study.
*Results Placement: Ensure all results-related content is presented in the appropriate sections for better organization and coherence.
*Discussion: Elaborate on the observed soil salinity trends and compare your findings with those of recent studies in the field.
*References: Update the references section by including recent publications to maintain academic rigor and relevance.
*Presentation: Enhance the quality of figures, tables, and the overall manuscript presentation to improve readability and visual appeal.
*Proofreading: Conduct thorough proofreading to ensure consistent language, grammar, and formatting throughout the manuscript.

Reviewer 1 ·

Basic reporting

1. Title
Consider making the title more specific and engaging. For example:
"Evaluating Soil Salinity Dynamics Under Drip Irrigation in the Manas River Basin, Xinjiang: A Long-Term Analysis (1996–2019)"
2. Abstract
Clarify Objectives: Highlight the novelty of your study. For instance:
Specify what makes the salinity inversion models innovative.
Explain the significance of geostatistical methods and water-salt balance analysis.
Key Findings: Quantify improvements in soil salinization more precisely (e.g., reduction trends over specific years).
3. Introduction
Provide a clear research gap: While you mention scarce studies on soil salinity under drip irrigation, elaborate on the specific aspects missing in existing research (e.g., lack of temporal or spatial analysis, insufficient model-based predictions).
Contextualize the study area: Add details about why the Manas River Basin is unique (e.g., climatic conditions, scale of irrigation, socio-economic relevance).
Emphasize the global relevance of your findings. How can lessons from Xinjiang be applied to other regions?

5. Results
Present key findings visually. For example:
Maps showing spatial changes in salinity across different years.
Graphs or trend lines illustrating salinity dynamics over time.
Quantify impact: For example, highlight annual salinization reduction rates or differences in soil salinity between irrigated and non-irrigated zones.
Discuss the threshold for irrigation and drainage with more specifics—e.g., define thresholds numerically or explain how they vary with groundwater depth.
6. Discussion
Compare results with prior research. How do your findings align with or differ from similar studies globally?
Explore mechanisms: Why does intensive drip irrigation inhibit salinization? Is it due to better salt leaching or groundwater control?
Highlight implications for policy and practice:
Suggest practical recommendations for irrigation management.
Address scalability of your methods in other regions.
Discuss future research directions: For example, assessing the role of climate variability in salinity dynamics.

Experimental design

4. Methodology
Detail the models and methods:
Describe the "soil salinity inversion models" in detail. Which variables are used? How are they validated?
Explain how water-salt balance analysis and geostatistical tools are applied.
Include information on the data sources, sampling methodology, and spatial resolution.
Address limitations: Discuss potential sources of error in regression models, or uncertainties related to field data collection.

Validity of the findings

7. Conclusion
Reiterate key findings succinctly but avoid repeating the results.
Link the findings to broader themes like sustainability, climate resilience, and global food security.
8. Language and Grammar
Fix grammar and clarity issues, e.g.:
Replace "ûrst" with "first" and "signiûcantly" with "significantly".
Rephrase awkward sentences such as "Drip irrigation technology is one of the main reasons for land salinization and low crop yield..."
Revised: "While drip irrigation technology has significantly improved water use efficiency, its improper use has been linked to land salinization and reduced crop yields."

Additional comments

9. References
Ensure the paper includes recent and region-specific references, as well as globally relevant studies on drip irrigation and salinization.
Consider citing landmark works in soil science, salinity modeling, and sustainable agriculture.
10. Broader Relevance
Highlight how the findings align with the UN Sustainable Development Goals (SDGs), particularly SDG 2 (Zero Hunger) and SDG 15 (Life on Land).
Discuss potential for technology transfer to similar regions facing salinity issues (e.g., arid regions in Central Asia, North Africa, or the Middle East).

Reviewer 2 ·

Basic reporting

- The paper lacks clarity in structure and organization. Ensure all sections (Abstract, Introduction, Methods, Results, and Discussion) are clearly delineated and logically connected.

- The abstract needs to be rewritten to include all essential components: study context, problem statement, methodology, key findings, and significance.

- Keywords should accurately reflect the study's content. Include terms such as "remote sensing," "soil salinity," specific sensors, and spectral indices to improve discoverability.

- The introduction requires significant improvement. Add a clear overview of remote sensing data types (e.g., optical, thermal, microwave) and their relevance to soil salinity mapping. Cite recent and relevant studies to contextualize your research.

- Update the literature review to include recent publications (within the last years) on soil salinity monitoring using remote sensing techniques and advanced models.

Experimental design

1. Methodology Presentation

• Add a flowchart summarizing the research methodology to improve clarity and flow. Clearly outline each step, from data acquisition to analysis and validation.
• Justify the choice of the equation used to calculate salt concentration. Compare it with alternative methods and provide references to support your decision.
• Clarify the need for supervised classification in a study that focuses on spectral indices. Explain its role and relevance in the workflow.

2. Remote Sensing Data Acquisition and Preprocessing

• Expand this section to include detailed descriptions of preprocessing steps, such as radiometric calibration, atmospheric correction, and geometric correction.
• Rename the section to "Remote Sensing Data Acquisition and Preprocessing" for better clarity.

3. Spectral Indices Selection

• Provide a robust justification for the choice of spectral indices. For instance:
• Explain why NDVI, which is vegetation-focused, was selected for soil salinity mapping.
• Discuss why SWIR-based indices, known for their sensitivity to soil salinity, were not prioritized.
• Address the correlation analysis of spectral indices with Soil Salt Concentration (SCC), and move this content to the Results section.

4. Model Development and Validation

• Clearly define the variables used in the soil salinity model and describe their significance.
• Provide a detailed explanation of the calibration and validation stages:
• Specify the datasets used for training and testing the model.
• Include the performance metrics (e.g., RMSE, R-squared) used to assess the model's accuracy.

Validity of the findings

1. Results Interpretation

The observed trend of decreasing soil salinity concentration (1996-2019) requires further explanation:
- Discuss potential drivers, such as improved land management practices, irrigation techniques, or climatic changes.
- Consider methodological limitations that might influence this trend and acknowledge any uncertainties.

2. Literature Integration

- Incorporate recent studies to validate your findings and strengthen the discussion. Highlight relevant research on soil salinity trends and remote sensing applications.
- Compare your results with similar studies to demonstrate consistency or discrepancies and provide a clear interpretation.

Additional comments

- Ensure the paper adheres to standard academic formatting and referencing guidelines.

- Address inconsistencies in the presentation of results and move any misplaced content (e.g., SCC correlation) to the appropriate section.

- Improve the quality of figures and tables by providing clear captions, units, and legends.

- Proofread the manuscript to correct any grammatical or typographical errors.

Reviewer 3 ·

Basic reporting

Overall paper is written in clear and in professional English language with the provision of sufficient background knowledge, enough literature review, good illustrated figures and tables etc. However some suggestions are given in separate file to be considered.

Experimental design

Study experiments are well designed however gaps can be filled considering the minor suggestion attached file.

Validity of the findings

This study is based on older methods with the lake of novelty but still have sound knowledge.

Additional comments

This paper offers a valuable contribution to soil salinity research through its integration of advanced techniques and long-term data. However, addressing the methodological gaps, improving clarity and organization, and strengthening actionable recommendations would significantly enhance its impact and utility for both scientific and practical purposes.

Annotated reviews are not available for download in order to protect the identity of reviewers who chose to remain anonymous.

---

## Round 0.2 · accepted · Accept

Authors have addressed all the comments. The manuscript can be accepted for the publication.

Reviewer 1 ·

Basic reporting

The authors have carefully considered all my comments therefore I recommend accepting the paper.

Experimental design

The authors have carefully considered all my comments therefore I recommend accepting the paper.

Validity of the findings

The authors have carefully considered all my comments therefore I recommend accepting the paper.

Reviewer 3 ·

Basic reporting

no comment

Experimental design

no comment

Validity of the findings

no comment

Additional comments

Authors included all the suggestions those were recommended in previous revision.